# A synthetic method to assay polycystin channel biophysics

**Megan Larmore[1], Orhi Esarte Palomero[1], Neha Kamat[2,3], Paul G DeCaen[1,4]***

[1]Department of Pharmacology, Feinberg School of Medicine, Northwestern University, Chicago, United States; [2]Department of Biomedical Engineering, McCormick School of Engineering and Applied Science, Northwestern University, Evanston, United States; [3]Center for Synthetic Biology, Northwestern University, Evanston, United States; [4]Chemistry of Life Processes Institute, Northwestern University, Evanston, United States

## eLife Assessment

The authors have developed a **valuable** approach that employs cell-free expression to reconstitute ion channels into giant unilamellar vesicles for biophysical characterisation. The work is **convincing** and will be of particular interest to those studying ion channels that primarily occur in organelles and are therefore not amenable to be studied by more traditional methods.

*For correspondence:
paul.decaen@northwestern.edu

**Competing interest:** The authors declare that no competing interests exist.

**Abstract** Ion channels are biological transistors that control ionic flux across cell membranes to regulate electrical transmission and signal transduction. They are found in all biological membranes and their conductive state kinetics are frequently disrupted in human diseases. Organelle ion channels are among the most resistant to functional and pharmacological interrogation. Traditional channel protein reconstitution methods rely upon exogenous expression and/or purification from endogenous cellular sources which are frequently contaminated by resident ionophores. Here, we describe a fully synthetic method to assay functional properties of polycystin channels that natively traffic to primary cilia and endoplasmic reticulum organelles. Using this method, we characterize their oligomeric assembly, membrane integration, orientation, and conductance while comparing these results to their endogenous channel properties. Outcomes define a novel synthetic approach that can be applied broadly to investigate channels resistant to biophysical analysis and pharmacological characterization.

## Introduction

Ion channels are pore-forming integral transmembrane proteins essential for all cellular lifeforms (*Hille, 1978*; *Lee et al., 2014*). At the plasma membrane, ion channels are responsible for generating long-range bioelectric conduction in excitable eukaryotic cells (e.g. neurons), and within prokaryotic colonies (e.g. bacteria biofilms), and filamentous colonies of archaea (*Yang et al., 2021*; *Prindle et al., 2015*; *Goaillard and Marder, 2021*). Here, they contribute to fundamental vital cell processes including division, signal transduction, and ionic homeostasis (*Hille, 1978*). In eukaryotic organelle membranes, ion channels are integral to a wide range of functions including energy production (mitochondria), and the maintenance of defining compartmental conditions such as $Ca^{2+}$ gradients (endoplasmic reticulum), pH (lysosome), and redox states (peroxisome) (*Stutzmann and Mattson, 2011*; *Smith and Aitchison, 2013*; *Li et al., 2019*). The conductive states of channels are precisely controlled by their molecular conformations which are unique among their phylogenetic families and subfamilies (*Yu et al., 2005*). More than 400 human genes encode ion channel subunits, many of which are

altered by variants that impact organ function and development (*George, 2014a*; *George, 2014b*; *Clare, 2010*). These so-called 'Channelopathies' manifest in various human diseases such as cardiac arrhythmias, neurological conditions, and cystic kidney diseases (*Lieve and Wilde, 2015*; *Kass, 2005*). Besides their association with disease-causing variants, channels are important therapeutic targets for treating various pathophysiologies and represent the second largest target among the existing FDA-approved drug portfolio (*Overington et al., 2006*; *Santos et al., 2017*; *Kaczorowski et al., 2008*). However, many members of the so-called 'dark genome' of understudied proteins encode putative ion channels that are implicated in human disease but remain resistant to functional characterization (*Oprea, 2019*). Furthermore, channels which localize to cellular compartments in low quantities present a significant challenge to assay for drug screening purposes (*McGivern and Ding, 2020*). Thus, these observations warrant the present investigation of a novel methodological approach to characterize ion channel biophysics and pharmacology.

Voltage-clamp electrophysiology incarnated in either planar or glass electrode designs provide direct, real-time and the highest available fidelity measurements of the ion channel conductive states. In the on-cell or inside-out configurations, their opening and closing conformational changes (i.e. gating) are captured as stochastic single-channel currents after forming high-resistance seals (>10 MΩ) at the interface of the electrode against a biological membrane (*Neher and Sakmann, 1976*). This conventional electrophysiology technique is commonly used to characterize the properties of plasma membrane channels and typically involves either recording from an endogenous cell source or from a cell line expressing the channel transgene (*Neher, 1992*; *Hamill et al., 1981*). However, capturing the biophysical properties of organelle channels and those from bacteria can be mired by electrode inaccessibility to inner membranes. As a work around, investigators have employed various channel reconstitution methods which typically consist of several steps (*Morera et al., 2007*; *Leptihn et al., 2011*). First, the channel of interest is expressed and immunopurified from a biological cell source, then it is reconstituted into a synthetic or into biologically derived lipid bilayers or vesicles. While there are many examples where heterologous approaches have faithfully reproduced the functional properties of channels from native sources, these preparations are frequently contaminated by endogenous ionophores from the host cell lines, even after purification (*Varghese et al., 2006*; *Pablo et al., 2017*).

To address this, we have developed a completely synthetic method to assay ion channel biophysics. The approach combines advances in cell-free protein expression (CFE) and reconstitution of the synthetic channel protein into giant unilamellar vesicles (GUV) where their single-channel properties can be measured using voltage-clamp electrophysiology (*Shimizu and Ueda, 2010*; *Kuruma et al., 2005*; *Jacobs and Kamat, 2022*). CFE is a form of in vitro protein synthesis, utilizing purified cellular machinery (ribosomes, tRNAs, enzymes, cofactors, etc.) needed to directly transcribe and translate user-supplied DNA plasmid encoding an ion channel. The unmodified channel proteins are reconstituted into GUVs— cell-sized model membrane systems derived from electrolysis of synthetic lipid mixtures. To validate this method, we characterized PKD2 and PKD2L1, two members of the polycystin subfamily of transient receptor potential (TRP) ion channels. PKD2 and PKD2L1 are highly homologous subunits with six transmembrane segments and shared overall protein folding when structurally assembled as homotetrameric channels (*Shen et al., 2016*; *Grieben et al., 2017*; *Wang et al., 2020*; *Wilkes et al., 2017*; *Hulse et al., 2018*; *Su et al., 2018*). Polycystin subunits can also form heteromeric ion channel complexes with several members of the TRP family and traffic to the primary cilia and endoplasmic reticulum organelle membranes (*Esarte Palomero et al., 2023*). Both features present challenges for experimentalists to functionally characterize these unique channel properties within their endogenous cell membranes. The medical and biological importance of polycystins is underscored by PKD2 variants associated with autosomal dominant polycystic kidney disease, and this channel's role in fertility and conferring right–left symmetry in embryonic development (*Wu et al., 1998*; *Gao et al., 2003*; *Tanaka et al., 2023*). While PKD2L1 variants have yet to be linked to human disease, its loss of expression results in epilepsy susceptibility and autism-like features in mice (*Vien et al., 2023*; *Yao et al., 2016*). In this report, we stepwise express and confirm protein expression of polycystin channels using the CFE method; reconstitute channel protein in GUVs containing distinct lipid components; assess correct membrane orientation using self-labeling saturated N-heterocyclic building blocks (SNAP) protein chemistry and evaluate channel properties using electrophysiology (*Vo et al., 2013*). Outcomes define a novel reductionist and generalizable approach to study ion channels resistant to biophysical characterization.

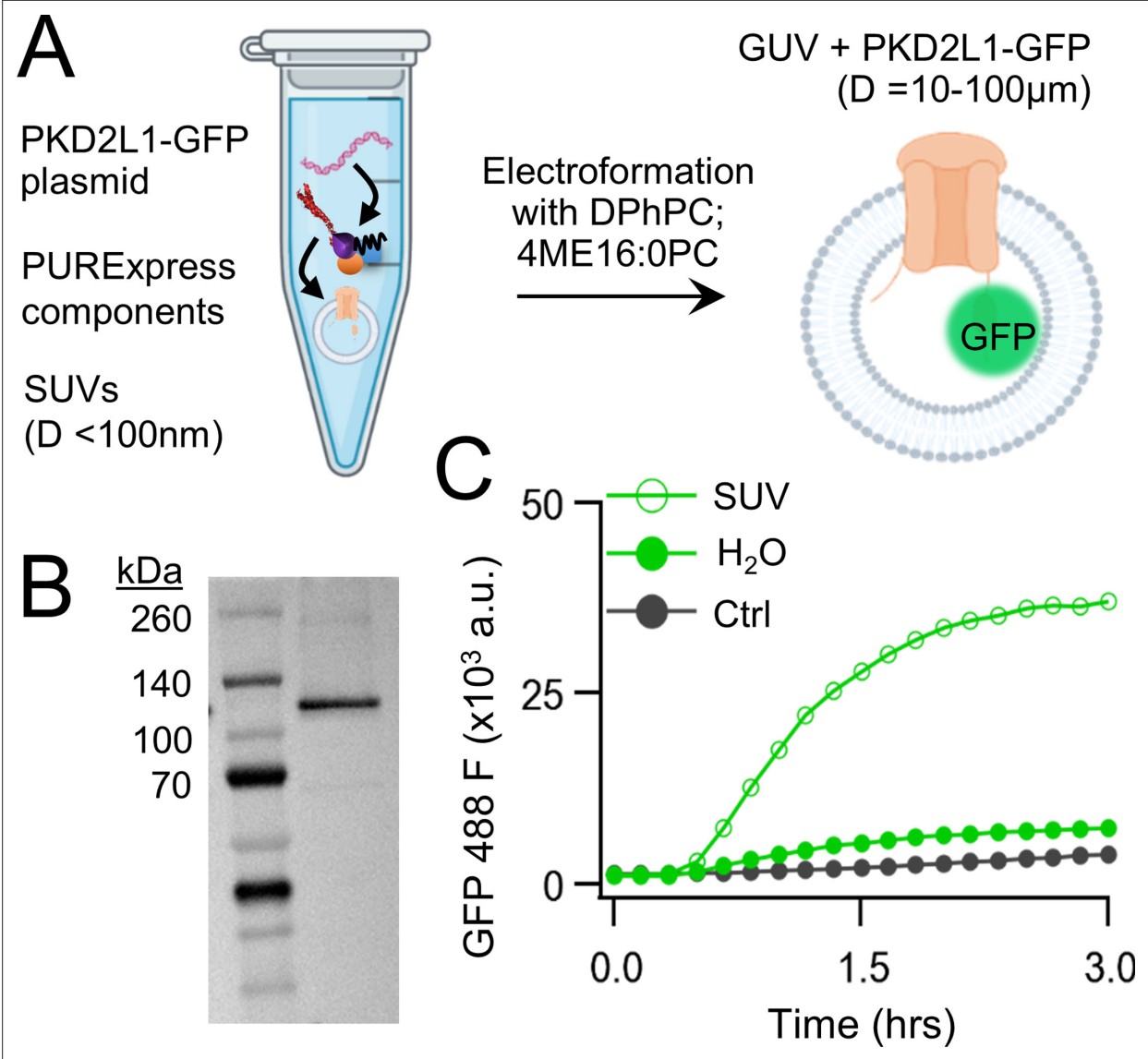

**Figure 1.** Cell-free expression of synthetic PKD2L1 protein and incorporation into lipid vesicles. (**A**) Schematic of cell-free protein expression into synthetic lipid vesicles and subsequent electroformation with Vesicle Prep Pro (Nanion). (**B**) Full-length PKD2L1-GFP protein detected by western blot after cell-free expression into vesicles. (**C**) Monitored fluorescence over time of cell-free expressed PKD2L1-GFP and a non-fluorescent control plasmid produced in the presence or absence of lipid vesicles.

The online version of this article includes the following source data and figure supplement(s) for figure 1:

**Source data 1.** Original file for western blot analysis displayed in *Figure 1B*.

**Source data 2.** PDF file containing original western blot for *Figure 1B*, indicating the relevant bands and treatments.

**Figure supplement 1.** Quantification, identification, and assembly of cell-free synthesized polycystin channels.

## Results and discussion

We began by carrying out CFE in vitro synthesis of PKD2L1 protein in the presence and absence of lipid vesicles. Plasmid DNA encoding human PKD2L1 with C-terminally tagged green fluorescent protein (PKD2L1-GFP) was added to the cell-free expression components (PURExpress, New England Biolabs) and induced protein translation by heat (*Figure 1A*, see methods). Each reaction produced 8 ± 3.5 ng/μl of synthetic channel protein as estimated by a standardized GFP absorbance assay (*Figure 1—figure supplement 1A*). The synthetically derived channel protein identity was confirmed using two methods. First, by western blot of the cell-free reaction where the PKD2L1-GFP protein was sodium dodecyl sulfate–polyacrylamide gel electrophoresis (SDS–PAGE) gel separated from the

reactants and detected by an anti-GFP monoclonal antibody (*Figure 1B*). Second, the PKD2L1 protein was confirmed by mass spectroscopy with 46% coverage (*Figure 1—figure supplement 1B, C*). Since polycystins are transmembrane proteins, we hypothesize that channel membrane incorporation would be facilitated by including lipid substrates into the CFE reaction. Thus, we compared channel synthesis in the presence or absence of small unilamellar vesicles (SUVs) comprised of 1,2-diphytanoyl-sn-glycero-phosphocholine (DPhPC; 4ME16:0PC) and cholesterol (*Figure 1A*). We monitored PKD2L1-GFP protein production over time through fluorescence which is dependent on complete channel translation and correct GFP folding (*Müller-Lucks et al., 2012*). We observed a dramatic increase in fluorescence output which plateaued after 3 hr when CFEs reactions were supplemented with SUVs— an effect not observed in water-supplement ($H_2O$) control reactions (*Figure 1C*; *Klammt et al., 2006*). Minimal changes in fluorescence were detected when a control plasmid (Ctrl) encoding a non-fluorescent protein (dihydrofolate reductase) was used in the reaction. Polycystin channels function as tetramers, thus we examined CFE-derived PKD2L1 oligomeric assembly in SUVs using fluorescence-detection size-exclusion chromatography (FSEC). As controls, we tested recombinant (cell-derived) GFP and GFP-tagged polycystin proteins which produced monodispersed peaks in the fluorescent signal which provided a reference for their respective elution times off the column (*Figure 1—figure supplement 1D*). Although not as robust, the fluorescence signal from CFE + SUV-derived PKD2L1 protein produced a symmetrical peak at the expected elution time of channel tetramers, along with fractions which may correspond to unassembled protomers (*Figure 1—figure supplement 1E*). Taken together, these results demonstrate the feasibility of synthesizing full-length polycystin channels using the cell-free expression system, and the enhancement of transmembrane protein synthesis and channel assembly through lipid vesicle incorporation during these reactions.

One caveat of membrane protein vesicle reconstitution is the potential for misorientation after lipid integration. To assay channel orientation, we synthesized PKD2L1 with a C-terminal SNAP-tag fusion protein (PKD2L1-SNAP) in SUVs then electroformed them into GUVs for the assay. The SNAP-tag specifically reacts with fluorescent $O_2$-benzylcytosine derivatives and, depending on the derivatives membrane permeability, will react with lipid integrated proteins based on the tag's accessibility (*Figure 2—figure supplement 1*). GUVs containing CFE synthesized PKD2L1-SNAP were formed by electroformation after passing current through indium tin oxide slides treated with dried SUV-channel protein mixture (*Boban et al., 2021*). We then added two SNAP fluorescent derivatives, membrane permeable SNAP-Cell Oregon Green (Cell488) and membrane impermeable SNAP-Surface Alexa Fluor 647 (Surface647), to monitor orientation-dependent membrane protein reactivity with the SNAP label (*Figure 2A*, *Figure 2—figure supplement 1*). We hypothesized there will be two fluorescent outcomes. First, if all channels orient correctly, then we should see only Cell488 fluorescence with no Surface647 at the membrane (*Figure 2A*). Second, if the channels are in opposite or in mixed orientations, then we expect to see dual fluorescence of Cell488 and Surface647 (*Figure 2B*). We imaged over 60 GUVs and found 38.5% of the vesicles exhibited sole Cell488 fluorescence—indicating their correct channel orientation within this population (*Figure 2B, C*). Importantly, none of the vesicle membranes labeled with both Cell488 and Surface647 while retaining a clear lumen—suggesting that the population of GUVs containing misoriented channels was nominal. In some cases, vesicles can encapsulate dye through compromised integrity or mechanisms other than membrane permeability. This is apparent when fluorescence can be seen in the vesicle lumen, rather than on the membrane (*Figure 2C*). While the encapsulated fluorescent vesicles account for most of the vesicles imaged, there were no vesicles seen with Cell488 and Surface647 fluorescence at the membrane with a clear lumen (*Figure 2B, C*). Based on these results, we conclude that our cell-free synthesized PKD2L1 channels are successfully reconstituted in GUVs in the correct orientation, and this preparation is suitable to assay PKD2L1 channels using electrophysiology.

Native and transgene expressed PKD2L1 channels conduct monovalent cations, thus we established symmetric potassium ion ($K^+$) recording conditions to measure synthetic polycystin currents from GUVs. GUVs containing PKD2L1 were voltage clamped using 4–7 MΩ resistance ($R$) glass electrodes in the inside-out membrane patch configuration (*Figure 3A*). Single-channel events were frequently observed while establishing high-resistance seals ($R > 10$ GΩ); however, the majority of PKD2L1 GUV patches were unstable during voltage steps and these results were excluded from the final analysis. We hypothesize that patch instability and low seal resistance likely results from over-incorporation of PKD2L1 tetramers into the GUV membranes, as supported by the FSEC data. In addition, membrane

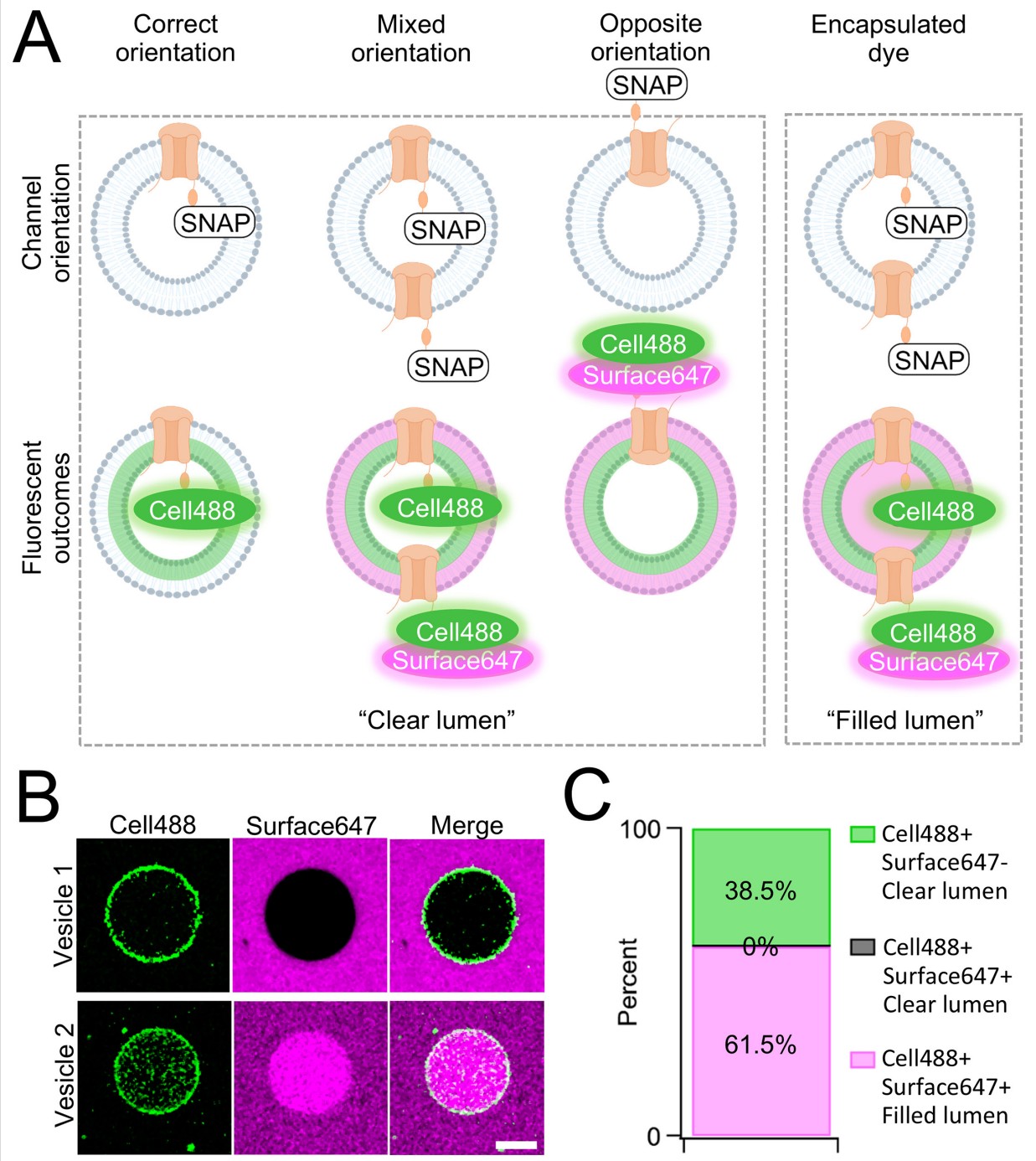

**Figure 2.** Orientation of synthetic PKD2L1 channels in vesicles. (**A**) Schematic of possible ion channel orientation outcomes from PKD2L1 cell-free expression (top) and hypothesized fluorescence results when Cell488 and Surface647 added (bottom). (**B**) Fluorescent confocal images from the SNAP-tagged vesicles. The scale bar represents 10 μm for all images. (**C**) Vesicle percentage depicts the percent of vesicles with each fluorescent output (*N* = 65 vesicles).

The online version of this article includes the following figure supplement(s) for figure 2:

**Figure supplement 1.** The SNAP-fluorescence approach to assess polycystin protein orientation in giant unilamellar vesicle (GUV).

instability was not observed from empty GUV recordings, suggesting that opening of incorporated CFE synthesized polycystins is likely responsible for patch destabilization. From the stable recordings, two magnitudes of single channels were readily observed from GUVs containing PKD2L1, suggesting unique full and sub-conductance states (*Figure 3B, C* and *Table 1*). In some recordings, only one

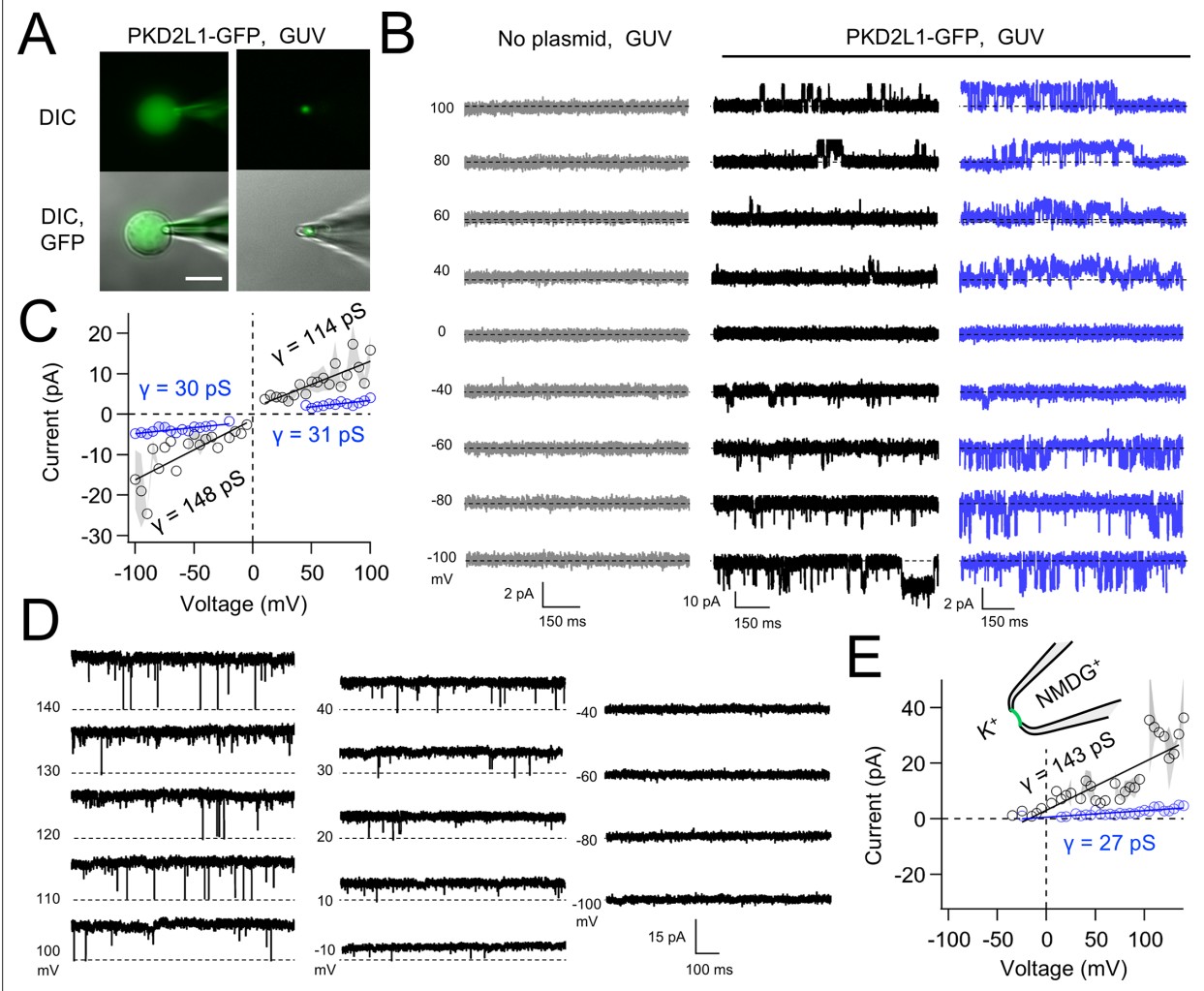

**Figure 3.** Synthetic PKD2L1 channels are functional in artificial membranes. (**A**) Images of voltage-clamped giant unilamellar vesicles (GUVs) with incorporated PKD2L1-GFP channels. *Left*, establishing high-resistance seals in the on-cell patch configuration. *Right*, transitioning to the inside-our patch configuration. Scale bar = 20 μm. (**B**) Example unitary single-channel current records from GUVs reconstituted with or without PKD2L1 protein. Vesicles were patched using the symmetrical 150 mM $K^+$ conditions (see methods) and PKD2L1 single-channel current producing full and sub-conductances are colored black and blue, respectively. (**C**) Average single-channel current amplitudes. Conductance ($\gamma$) estimated by fitting the average single-channel currents to a linear equation. Error (gray) indicates SEM from $N$ = 3–8 GUVs. Several replicates lacked single-channel openings at all test potentials. (**D**) PKD2L1 single-channel current recorded using asymmetric cationic solutions, with 150 mM $K^+$ in the bath and 150 mM $NMDG^+$ in the pipette. (**E**) Resulting average single-channel current amplitudes where no inward single-channel current was detected ($N$ = 3–4 GUVs).

The online version of this article includes the following figure supplement(s) for figure 3:

**Figure supplement 1.** Synthetic polycystin channels exhibit full and sub-conductive states in giant unilamellar vesicles (GUVs).

conductance predominates, which can be estimated from recordings from individual GUVs (*Figure 3— figure supplement 1A, B*). While in other recordings, both full (*FC*) and sub-conductive (*SC*) states can be identified by histogram analysis of the unitary current (*Figure 3—figure supplement 1C, D*). Importantly, no single-channel events were observed from GUVs ($N$ = 11) derived from CFE reactions with the empty plasmid—indicating that the measured conductance is not due to contaminates from lipid or cell-free reagents (*Figure 3B*). To assess the selectivity of the synthetic PKD2L1 pore, we substituted the pipette $K^+$ charge carrier for methyl-D-glucamine ions ($NMDG^+$). We did not observe any outward single-channel currents (i.e. toward the bath), indicating the large cation was not permeable through PKD2L1 which is consistent with previous reports (*Figure 3D, E*; *DeCaen et al., 2016*; *Ng et al., 2019*). To determine the feasibility of using this approach to assess the function of other polycystin channels, we followed the same steps to assay PKD2 channel biophysics. As observed in

**Table 1.** Conductance properties of polycystins measured from giant unilamellar vesicle (GUV) and cilia membranes.

| Polycystin channel and membrane context | Major K⁺ g (pS) ± SD | | Minor K⁺ g (pS) ± SD | |
|---|---|---|---|---|
| | Inward | Outward | Inward | Outward |
| PKD2 (GUV membrane, cell-free expression) | 282 ± 38 | 153 ± 32 | 23 ± 4 | 21 ± 3 |
| PKD2 (primary cilia membrane, endogenous murine inner medullary collecting duct [IMCD] cell line) | 144 ± 9 | 110 ± 6 | 46 ± 4 | 34 ± 4 |
| PKD2L1 (GUV membrane, cell-free expression) | 148 ± 29 | 114 ± 28 | 30 ± 6 | 31 ± 7 |
| PKD2L1 (primary cilia membrane, endogenous hippocampal neurons) | 165 ± 10 | 113 ± 6 | 40 ± 4 | 29 ± 3 |

our PKD2L1 results, unitary single-channel currents of synthetic PKD2 channels reconstituted in GUVs yield sub- and full conductances, which were terminated by substitution of K⁺ with NMDG⁺ in the electrode (*Figure 4A–C*; *Figure 3—figure supplement 1E–H*). To compare the properties of the synthetic and biologically derived channels, we recorded native PKD2L1 and PKD2 channel single-channel currents from the primary cilia membranes of hippocampal neurons and inner medullary collecting duct (IMCD) cell line, respectively (*Figure 4—figure supplement 1*; *Vien et al., 2023*; *Kleene and Kleene, 2017*; *Liu et al., 2018*). Like the synthetic PKD2 and PKD2L1 channels, native polycystins produced sub- and full K⁺ conductances with inward currents having the greater magnitudes. Here,

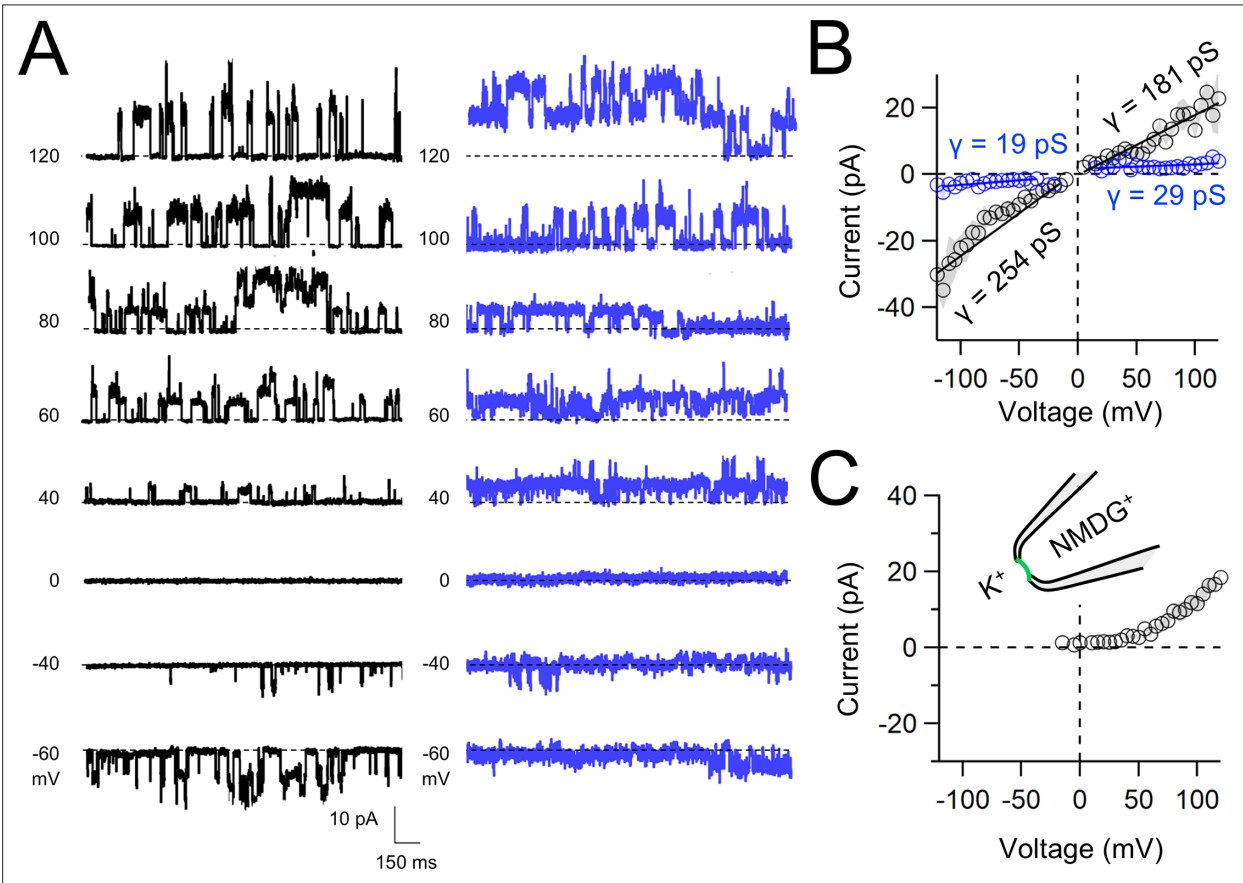

**Figure 4.** Functional synthetic PKD2 channels in artificial membranes. (**A**) Example unitary single-channel current records from giant unilamellar vesicles (GUVs) reconstituted with PKD2 protein in symmetrical 150 mM K⁺ conditions producing full (black traces) and sub-conductances (blue traces), respectively. (**B, C**) Average single-channel current amplitudes recorded using K⁺ or NMDG⁺ in the recording electrode solution. Conductance ($\gamma$) estimated by fitting the average single-channel currents to a linear equation. Error (gray) indicate SEM from K⁺ (N = 3–12 GUVs) and NMDG⁺ (N = 2–5 GUVs) conditions.

The online version of this article includes the following figure supplement(s) for figure 4:

**Figure supplement 1.** Native polycystin channels measured from primary cilia membranes exhibit full and sub-conductance states.

the synthetic PKD2L1 GUV conductance approximates the native full and sub-conductances recorded from hippocampal primary cilia membranes cultured from neonatal mice (ARL13B-EGFP$^{tg}$). However, the PKD2 K$^+$ conductance magnitudes recorded from IMCD cilia were significantly smaller than those assayed using the CFE–GUV synthetic system (*Table 1*). These differences might arise from the lack of post-translational modifications (e.g. phosphorylation and *N*-glycosylation) to the synthetic PKD2 peptides, which are normally found in biologically derived channels (*Su et al., 2018*; *Cai et al., 1999*; *Newby et al., 2002*). In addition, the GUVs are comprised of synthetic lipids which does not reflect the composition of organelle (cilia or ER) membranes of the cell (*Nakatsu, 2015*). Thus, while retaining the native ion selectivity and ion channel functionality despite their cell-free origin, synthetic PKD2 has different conductance magnitudes compared to cell-derived channels, which presents a limitation of using this approach to recapitulating physiological channel functions.

In summary, we have established a synthetic approach to assay ion channel biophysics using two polycystin members to validate our method. Previously CFE has been used to study membrane protein integration, drug delivery, and the study of actin dynamics (*Jacobs and Kamat, 2022*; *Noireaux and Liu, 2020*; *Göpfrich et al., 2019*). The novelty of our approach rests with the adaptation of CFE-derived channels and their GUV reconstitution to carry out single-channel electrophysiology experiments. The described method represents a highly reductionist approach to assay channel function which can be generalized to other channels resistant to characterization using traditional electrophysiology approaches. Furthermore, the CFE–GUV electrophysiology method can be leveraged for future inquiry into lipid-channel regulation and effects of channel subunit composition. PKD2-related protomers form heteromeric complexes (e.g. PKD1–PKD2; PKD1L1–PKD2L1) with PKD1-related polycystins which are notoriously difficult to assay using traditional electrophysiology techniques (as reviewed) (*Esarte Palomero et al., 2023*). Using the CFE method to co-synthesize and GUV reconstitute these subunits presents a potential avenue to assay their function with patch clamp recordings. The approach may be further developed into high throughput drug screening assays using cation reporters (e.g. Fura derivatives) or multi-well planer electrophysiology instruments (*Zhou et al., 2021*; *Yu et al., 2016*). With our pipette patch electrodes, we observed considerable instability of our high resistant seals when the bath solutions were exchanged. Thus, future work could explore alternative membrane compositions (e.g. additional cholesterol) and stabilizing cationic conditions (internal CsF) to mitigate this effect. The folding and membrane integration of many ion channels require their association with stabilizing chaperone proteins produced in cells (*Li et al., 2017*; *Chen et al., 2023*; *Bai et al., 2018*). Thus, while the co-expression of chaperones with these channels using the CFS + GUV system is likely required for their functionality, the approach may also be leveraged to study chaperone-assisted folding in vitro.

## Methods

### Protein production

Cell-free protein production was performed with PURExpress In Vitro Protein Synthesis Kit from New England Biolabs, Inc (Ipswich, MA, USA). Both PKD2 and PKD2L1 are in a pET19b plasmid under T7 promoter. We utilized the manufacturer's protocol with 1 mg target DNA and a maximum reaction volume of 30 µl. When appropriate, we added substituted diH$_2$O for SUVs. The reactions were placed in 37°C water bath or heated plate reader between 2 and 3 hr and then placed at 4°C for storage.

### Vesicle formation

Lipids DPhPC (4ME16:0PC) and cholesterol (ovine) were obtained from Avanti Polar Lipids (Alabaster, AL, USA) and mixed in chloroform to the desired mol percentage, 95% DPhPC and 5% cholesterol. SUVs were formed following the previously described (*Varghese et al., 2006*). Briefly, lipids were reconstituted in chloroform in a glass vial and the chloroform was evaporated until a thin lipid layer is deposited on the bottom of the glass vial. The lipid layer is then placed under vacuum, –23 inhg, for >4 hr. Lipids are then rehydrated in 1 ml of diH$_2$O overnight at 60°C. The following day, lipids are vortexed and then passed through a 100-nm polycarbonate filter with the Mini-Extruder (Avanti Polar Lipids, Alabaster, AL, USA) seven times and stored at 4°C for 2 weeks. SUVs with or without channel incorporated are dried onto indium tin oxide-coated glass slides from Nanion Technologies (Munich, Germany). The dried slides are placed on the Vesicle Prep Pro (Nanion Technologies) with a rubber

o-ring and 300 mM sucrose. GUVs are formed using the standard program. GUVs are electroformed and used for electrophysiology experiments the same day. SUVs with channel incorporated are stored at 4°C for 3 days.

## Monitoring fluorescence and cell-free protein synthesis quantification

We monitored fluorescent folding with PKD2L1 C-terminally tagged GFP during PURExpress reaction in the presence and absence of SUVs. GFP fluorescence was monitored every 10 min for 3 hr at 37°C on the BioTek Cytation5 Imaging Reader (Agilent, Santa Clara, CA, USA). Control plasmid was the PURExpress Control DHFR Plasmid (NEB, Ipswich, MA, USA) with no fluorescent tag. GFP standard curve was created from dilutions of Aequorea Victoria GFP His-tag recombinant protein (Thermo Fisher Scientific, Cat. No. A42613) and measured on the BioTek Cytation5 Imaging Reader. A linear regression (Igor Pro, WaveMetrics, Portland, OR, USA) was used to create a standard curve. Target protein fluorescent measurements were made after in vitro protein synthesis for 3 hr.

## Western blotting

Western blotting was performed on PKD2L1–GFP plasmid after PURExpress protein expression in the presence of SUVs. SUV and protein mixture were separated on SDS–PAGE gel, Novex Tris-Glycine mini protein gels, 4–20%, 1.0 mm, WedgeWell format (Thermo Fisher, Waltham, MA, USA). The SDS–PAGE was run with 10 µl Spectra Multicolor Broad Range Protein Ladder (Thermo Fisher Scientific, Cat. No. 26634). The gel was then transferred to Amersham Hybond P 0.45 PVDF blotting membrane (Cytiva, Cat. No. 10600029) and PKD2L1-GFP was detected with an anti eGFP monoclonal antibody (F56-6A1.2.3) (Invitrogen, Cat. No. MA1-952) diluted 1:1000 in TBS with 0.1% (vol/vol) Tween-20 and 5% (5/vol) milk overnight at 4°C. The goat anti-mouse AF555 secondary (Invitrogen, Cat. No. A32727) diluted 1:5000 in TBS with 0.1% Tween-20 and 5% milk was incubated for 1 hr at room temperature.

## SNAP staining

Channel orientation was visualized with N-terminally tagged PKD2L1 with SNAP Tag sequence (NEB, Ipswich, MA, USA). After PKD2L1-SNAP-tag production and incorporation into GUVs, two SPAP-tag substrates were added to the solution, cell permeable SNAP-Cell Oregon Green and the cell impermeable SNAP-Surface Alexa Fluor 647, according to the manufacturer's instructions. Images were collected on Nikon A1 confocal microscope and all images were analyzed with Nikon Elements (Melville, NY, USA). Briefly, regions of interests were manually outlined around the vesicle membrane. Then Pearson's correlation coefficients were measured for fluorescence overlap of the two substrates.

## Fluorescence-detection size-exclusion chromatography

As controls for the polycystin FSEC elution time, Aequorea Victoria GFP His-tag recombinant protein (Thermo Fisher Scientific, Cat. No. A42613) and polycystin channel protein was harvested from lysates of $0.5 \times 10^9$ HEK293T (ATCC, Cat. No. CRL-3216) PKD2$^{Null}$ cell lines (*Vien et al., 2020*) stably expressing PKD2-GFP and PKD2L1-GFP. Sythetic channel protein from three CFE reactions were synthesized in the presence of SUVs, as previously described. The SUVs were lysed using dodecyl β-D-maltoside (DDM) (GoldBio) and protein supinates collected after centrifugation for 20 min at 20,000 rpm. Samples were then diluted in 50–100 µl of SEC running buffer containing (ml) 150 mM NaCl, 25 mM N-2-hydroxyethylpiperazine-N'-2-ethanesulfonic acid (HEPES), 1 mM CaCl$_2$·6H$_2$O, 0.05% DDM, 0.005% cholesteryl hemisuccinate, pH 7. 50 µl of each sample were separated on an analytical size-exclusion column (Superose 6 5/150 GL; GE Healthcare) at 0.2 ml/min flow rate. Fluorescent proteins were detected (excitation: 480 nm, emission: 512 nm) using an RF-20Axs detector (Shimadzu, Japan).

## Isolation of primary hippocampal neurons and IMCD cell culture for cilia electrophysiology

All mice utilized in these procedures are housed in our AAALAC-approved Center for Comparative Medicine (CCM) at Northwestern University, Feinberg School of Medicine. NU Institutional Animal Care and Use Committee has approved this facility and was monitored by an Animal Care Supervisor as well as a veterinarian from the Division of Laboratory Animal Medicine (DLAM). All of those who handled animals and perform the approved protocols were properly trained prior to start of work

to ensure no animal discomfort. Mice were anesthetized using isoflurane and sacrificed by severing the spinal cord. Hippocampi were dissected from 3 to 6 ARL13B-EGFP[tg] mice (ages P0–P1, sex undetermined, background strain C57BL/6J) and digested in ~20 units of papain (LK003176, Worthington) and ~200 units of deoxyribonuclease (LK003172, Worthington) dissolved to basal medium eagle solution (B1522, Sigma) at 37°C for 25 min (*Vien et al., 2023*). Tissues were washed with beta-mercaptoethanol (BME) and triturated to release cells. Cells were centrifuged at 300 × *g* for 4 min and plated on polylysine-coated glass coverslip at 1–4 × 10⁵ density in basal medium eagle solution containing B27 supplement (17504044, Gibco), N-2 supplement (17502048, Gibco), 0.5% penicillin/streptomycin (15140148, Gibco), 5% fetal bovine serum, 5% horse serum (260500,Gibco), and GlutaMax (350500, Gibco). Hippocampal neurons were cultured for 2–7 days prior to conducting electrophysiology experiments. IMCD cells stably expressing ARL13B-EGFP cilia reporters were cultured in F12/DMEM media (Sigma) with 10% fetal bovine serum (Sigma) and 50 I.U./ml penicillin–streptomycin (30-2300, ATCC) antibiotic.

## Electrophysiology

All research chemicals used in the electrophysiology experiments were purchased from Millipore-Sigma. Single-channel recordings were recorded from primary cilia and GUV membranes. All GUV patch electrodes were made using borosilicate glass electrodes and were fire polished to resistances greater than 5–10 MΩ and primary cilia patch electrodes were polished to a resistance greater than 15 MΩ. Renal primary cilia PKD2 currents were recorded from mIMCD-3 (ATCC, Catalog No. CRL-2123) cells expressing the cilia reporter ARL13B-EGFP, as previously described (*Kleene and Kleene, 2017*; *Kleene and Kleene, 2012*; *DeCaen et al., 2013*; *Vien et al., 2020*; *Liu et al., 2017*). Primary cilia PKD2L1 currents were recorded from isolated neonatal hippocampal neurons from ARL13B-EGFP[tg] using previously described procedure (*Vien et al., 2023*). Recording solutions for mammalian culture consisted of symmetrical recording solutions with 150 mM KCl, 10 mM HEPES, and 300 mM glucose, unless the charge carrier was changed when mentioned. Recordings were collected in voltage clamp with voltage steps from −100 to +100 mV and a holding potential of −40 mV with ClampEx v.11.2.0.59 (MolecularDevices, San Jose, CA, USA) using a Axopatch 200B amplifier. Recordings were digitized with the Digidata 1550B (MolecularDevices) at 25 kHz and low pass filtered at 5 kHz. Recordings were analyzed with ClampFit v11.2.0.59 (Molecular Devices, San Jose, CA, USA) and IGOR Pro 8.04 (WaveMetrics, Portland, OR, USA). As a predetermined criteria, data was excluded from analysis when seal resistance fell below 5 MΩ due insufficient voltage control of the patched membrane. Conductance was determined by determining the slope of the current–voltage relationship. Probability of open time was calculated by measuring the time at which a channel spends in an open confirmation divided by the total time in the voltage step.

## Materials availability statement

All CFE and mammalian cell expression constructs used in this study are available without restriction upon written request to the corresponding author.

## Acknowledgements

We acknowledge members of the Kamat and DeCaen labs for their constructive comments during the drafting of this manuscript. We thank Dr Alfred George for the use of his lipid electroformation equipment used to generate SUVs and GUVs. ML was supported by Northwestern University's (NU) molecular biophysics training grant (T32 GM008382) and the National Institute of Diabetes and Digestive and Kidney Diseases (NIDDK) of the National Institutes of Health kidney, urologic and hematologic (KUH) disease training grant (U2CDK129917). OEP was supported by the Ruth L Kirschstein National Research Service Award (NRSA) individual postdoctoral fellowship (F32DK137477-01A1) and NU KUH training grants (U2CDK129917 and TL1DK132769); PGD was supported by the NIH NIDDK grants R01 DK123463-01 and R01 DK131118-01.

# Additional information

## Funding

| Funder | Grant reference number | Author |
|---|---|---|
| National Institutes of Health | T32 GM008382 | Megan Larmore |
| National Institutes of Health | U2CDK129917 | Megan Larmore<br>Orhi Esarte Palomero |
| National Institutes of Health | TL1DK132769 | Orhi Esarte Palomero |
| National Institute of Diabetes and Digestive and Kidney Diseases | F32DK137477-01A1 | Orhi Esarte Palomero |
| National Institute of Diabetes and Digestive and Kidney Diseases | R01 DK123463-01 | Paul G DeCaen |
| National Institute of Diabetes and Digestive and Kidney Diseases | R01 DK131118-01 | Paul G DeCaen |

The funders had no role in study design, data collection, and interpretation, or the decision to submit the work for publication.

## Author contributions

Megan Larmore, Data curation, Formal analysis, Validation, Investigation, Visualization, Methodology, Writing – original draft; Orhi Esarte Palomero, Formal analysis, Investigation, Writing - review and editing; Neha Kamat, Conceptualization, Supervision; Paul G DeCaen, Conceptualization, Data curation, Supervision, Visualization, Writing – original draft, Project administration

## Author ORCIDs

Orhi Esarte Palomero (iD) https://orcid.org/0000-0003-2022-3140
Neha Kamat (iD) https://orcid.org/0000-0001-9362-6106
Paul G DeCaen (iD) https://orcid.org/0000-0001-8776-983X

Reviewer #1 (Public review): https://doi.org/10.7554/eLife.98534.3.sa1
Reviewer #2 (Public review): https://doi.org/10.7554/eLife.98534.3.sa2
Author response https://doi.org/10.7554/eLife.98534.3.sa3

# Additional files

## Supplementary files

• MDAR checklist

## Data availability

Data reported in this paper is deposited and available without restriction at the NU library ARCH (https://doi.org/10.21985/n2-4hs8-6j16).

The following dataset was generated:

| Author(s) | Year | Dataset title | Dataset URL | Database and Identifier |
|---|---|---|---|---|
| Larmore M, DeCaen P | 2024 | Larmore_et_al_2024_A synthetic method to assay polycystin ion channel biophysics | https://doi.org/10.21985/n2-4hs8-6j16 | Arch, 10.21985/n2-4hs8-6j16 |

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
