## [Editor Report · eLife Assessment]

The authors have developed a **valuable** approach that employs cell-free expression to reconstitute ion channels into giant unilamellar vesicles for biophysical characterisation. The work is **convincing** and will be of particular interest to those studying ion channels that primarily occur in organelles and are therefore not amenable to be studied by more traditional methods.

---

## [Referee Report · Reviewer #1 (Public review)]

Summary:

The authors have developed a valuable method based on a fully cell-free system to express a channel protein and integrated it into a membrane vesicle in order to characterize it biophysically. The study presents a useful alternative to study channels that are not amenable to be studied by more traditional methods.

Strengths:

The evidence supporting the claims of the authors is solid and convincing. The method will be of interest to researchers working on ionic channels, allowing to study a wide range of ion channel functions such as those involved in transport, interaction with lipids or pharmacology.

Weaknesses:

The inclusion of a mechanistic interpretation how the channel protein folds into a protomer or a tetramer to become functional into the membrane, would strengthen the study.

Comments on revised version:

In the revised version, the authors did not experimentally addressed how tetrameric or protomeric proteins are actually produced. However, they performed new experiments to assess the amount of tetramers that are being actually formed. They used a size-exclusion chromatography to conclude that the protomers and tetramers species of complexes are formed and assembled.

The authors have addressed most of my minor concerns and have modified or updated the manuscript following my recommendations, so I have no further comments.

---

## [Referee Report · Reviewer #2 (Public review)]

It is challenging to study the biophysical properties of organelle channels using conventional electrophysiology. The conventional reconstitution methods requires multiple steps and can be contaminated by endogenous ionophores from the host cell lines after purification. To overcome this challenge, in this manuscript, Larmore et al. described a fully synthetic method to assay the functional properties of the TRPP channel family. The TRPP channels are an important organelle ion channel family that natively traffic to primary cilia and ER organelles. The authors utilized cell-free protein expression and reconstitution of the synthetic channel protein into giant unilamellar vesicles (GUV), the single channel properties can be measured using voltage-clamp electrophysiology. Using this innovative method, the authors characterized their membrane integration, orientation, and conductance, comparing the results to those of endogenous channels. The manuscript is well-written and may present broad interest to the ion channel community studying organelle ion channels. Particularly because of the challenges of patching native cilia cells, the functional characterization is highly concentrated in very few labs. This method may provide an alternative approach to investigate other channels resistant to biophysical analysis and pharmacological characterization.

Comments on revised version:

The authors have addressed my concerns. This excellent method manuscript would benefit the study of organelle channels.

---

## [Author Response]

The following is the authors’ response to the original reviews.

**Public reviews:**

**Reviewer #1 (Public Review):**
Summary:The authors have developed a valuable method based on a fully cell-free system to express a channel protein and integrate it into a membrane vesicle in order to characterize it biophysically. The study presents a useful alternative to study channels that are not amenable to being studied by more traditional methods.Strengths:The evidence supporting the claims of the authors is solid and convincing. The method will be of interest to researchers working on ionic channels, allowing them to study a wide range of ion channel functions such as those involved in transport, interaction with lipids, or pharmacology.Weaknesses:The inclusion of a mechanistic interpretation of how the channel protein folds into a protomer or a tetramer to become functional in the membrane would strengthen the study.

Work from other labs has described key factors which can improve expression and artificial lipid integration of cellfree derived transmembrane proteins (PMIDs: 35520093, 29625253, 26270393) . However, a significant number of additional experiments would be needed to elucidate the exact biophysical properties governing channel assembly of synthetically derived polycystins. We carried out additional biochemical experiments to address these concerns (see new Figure 1— figure supplement 1 D, E). We used fluorescence-detection size-exclusion chromatography (FSEC) with the goal of understanding how much of the CFE-derived protomers are biochemically folding and assembly into functional tetramers upon incorporation into SUVs. When compared to protein recombinant sources from HEK cells, the production of assembled channels is less than 4% when using the CFE+SUV approach, an estimate based on the oligomer peak fluorescence. In the absence of chaperones found in cells, the assembly of synthetically derived protomers into tetramers is likely intrinsic to the chemical properties of the proteins, and the biophysical principles governing helical membrane protein when inserted into the lipid membrane (PMID:35133709). We have added our interpretation in lines 111-121.

**Reviewer #2 (Public Review):**
It is challenging to study the biophysical properties of organelle channels using conventional electrophysiology. The conventional reconstitution methods require multiple steps and can be contaminated by endogenous ionophores from the host cell lines after purification. To overcome this challenge, in this manuscript, Larmore et al. described a fully synthetic method to assay the functional properties of the TRPP channel family. The TRPP channels are an important organelle ion channel family that natively traffic to primary cilia and ER organelles. The authors utilized cell-free protein expression and reconstitution of the synthetic channel protein into giant unilamellar vesicles (GUV), the single channel properties can be measured using voltage-clamp electrophysiology. Using this innovative method, the authors characterized their membrane integration, orientation, and conductance, comparing the results to those of endogenous channels. The manuscript is well-written and may present broad interest to the ion channel community studying organelle ion channels. Particularly because of the challenges of patching native cilia cells, the functional characterization is highly concentrated in very few labs. This method may provide an alternative approach to investigate other channels resistant to biophysical analysis and pharmacological characterization.

Thank you for evaluating our manuscript.

**Recommendations for the authors:**

**Reviewer #1 (Recommendations For The Authors):**
(1) It would be useful to explain how the Polycystin protein is folded under the experimental conditions used. The expression data shown in Figure 1 Supplement 1B show different protein concentrations of protomer or tetramer. However, it is not described how each form is identified and distinguished. It is also important to mention in the manuscript that this method is only applicable to membrane channels that do not require chaperons for its folding and expression into the membrane. How is the tetramer mechanistically conformed? In line 184, it is stated that this method can be leveraged for studying the effects of channel subunit composition. Would this method allow the expression of two different subunit proteins in order to produce a heteromeric channel?

In Figure 1—figure supplement 1B, total fluorescence from the synthesized channel-GFP was measured. Protein concentration was calculated based on the linear regression of the GFP standards. Monomeric protein concentration was reported directly from total fluorescence. Tetrameric protein concentration was calculated by dividing the fluorescence by four, and subsequently calculating the concentration based off the GFP standards.

This is a good point. Based on your suggestion, we carried out additional biochemical experiments (see new Figure 1— figure supplement 1 D, E). We used fluorescence-detection size-exclusion chromatography (FSEC) with the goal of understanding how much of the CFE-derived protomers are biochemically folding and assembly into functional tetramers upon incorporation into SUVs. As controls we produced recombinant PKD2-GFP and PKD2L1GFP channels as elution time standards and to compare the relative production of tetrameric channels generated when using the two expression systems. The synthetically derived polycystin channels indeed produced tetramers and protomers, which supports feasibility of using this method to assay their functional properties. When compared to protein recombinant sources from HEK cells, the production of assembled channels is less than 4% when using the CFE+SUV approach, an estimate based on the oligomer peak fluorescence. We speculate that assembly of synthetically derived protomers into tetramers is likely intrinsic to the chemical properties of the proteins, and the biophysical principles governing helical membrane protein when inserted into the lipid membrane (PMID: 35133709). Although an interesting question, a systematic analysis of these channel-lipid interactions is beyond the scope of this eLife Report but can be addressed in future studies. The limitation of using this method to characterize channels which fold and membrane integrate without the aid of molecular chaperones is now stated in lines 201205. In principle, the CFE-GUV method can be deployed to co-express different subunits to produce heteromeric channels. We have modified the text lines 192-197 to be clearer on this point.

(2) The type of plasmid (and promoter) required for this methodology should be mentioned.

Added to the methods (lines 210-211). “PKD2 and PKD2L1 are in pET19b plasmid under T7 promoter.”

(3) Since this paper is methodological, it would be useful to have some information about the stability of the GUVs containing the synthetic channel. In Methods, it is stated that GUV vesicles are used on the same day (line 207). And in line 193 it says that the reactions (?) are placed at 4{degree sign}C for storage.

Restated in lines 226-228: GUVs are electroformed and used for electrophysiology the same day. SUVs with channel incorporated are stored at 4°C for 3 days.

(4) A comment reasoning why the PKD2 protein is more frequently incorporated into the membrane in comparison to PKD2L1 should be included. A brief description of the differences between these two proteins would also be helpful for the reader.

In terms of overall protein production and oligomeric assembly— more PKD2L1 channels are produced compared to PKD2 (see new Figure 1C, and Figure 1— figure supplement 1 D, E). In lines 149-155 we note single channel openings were frequently observed for the high expressing PKD2L1 channels, but this often resulted in patch instability. As a result, GUV patches with lower expressing PKD2-GFP channel were more stable and thus more successfully recorded from. We have revised the text to be clearer on this point.

(5) There are no methods for preparing hippocampal neurons or IMCD cells shown in Figure 4 Supplement 1. Instead, the method of mammalian cultures provided corresponds to HEK 293T cells.

This information has been added to lines 273-284.

(6) Minor:In Figure 2C, please include the actual % of the Cell488+Surface647+Clear lumen vesicles.

Added

Line 99, 108: Figures 1B and 1C are swapped. Please correct.

Corrected in figure and figure legends.

Line 108: misspelling: effect.

Done

Line 109: check sentence: verb is missing.

Sentence now reads “Minimal changes in fluorescence were detected when a control plasmid (Ctrl) encoding a non- fluorescent protein (dihyrofolate reductase) was used in the reaction.”

Line 145: recoding. Correct.

Recoding changed to recordings

Line 169: "from" is missing (recorded from MCD cilia).

Added

Line 169: In Table 1, the PKD2 K+ conductance magnitudes recorded from IMCD cilia were significantly smaller, not larger as stated, than those assayed using CFE-GUV system. Please correct.

Corrected

Line 180: "of" is missing (adaptation of CFE derived...).

Corrected

Line 182: "to" is missing (generalized to other channels).

Corrected

Line 193: "in" 4ºC, correct at.

Corrected

Line 197: replace "mole" for "mol".

Corrected

Line 207: are used "within the" same day.

Corrected

Line 210: c-terminally. C-should be capital letter.

Corrected

Line 231: n-terminally. N- should be capital letter.

Corrected

**Reviewer #2 (Recommendations For The Authors):**
The authors validated their method using PKD2 and PKD2L1 channels, demonstrating the potential of this approach. However, a few points merit further clarification or validation:(1) Stability of the protein vesicles for recording. The authors observed membrane instability during voltage transitions. It would be beneficial to discuss potential solutions to enhance stability.

In lines 197-202, we have added a discussion of potential solutions to enhance stability. CsF in the intracellular saline could be added to stabilize the GUV membranes. CsF is frequently added to stabilize whole cell membranes in HTS planer patch clamp recording. We did not explore this formulation because Cs+ would limit outward polycystin conductance. We also suggest but did not test altering the membrane formulation of GUVs with additional cholesterol to stabilize these recordings.

(2) Validation. Further discussion on how broadly this method can be applied to other channels would strengthen the manuscript.

We have included further discussion on this point in lines 190-206.

(3) Protein production estimated by a standard GFP absorbance assay. The estimation of protein production using GFP absorption may be affected by improperly folded protein. Additional validation methods could be considered.

C-terminal GFP fluorescence has been widely used in expression systems to designate proper folding of the target protein upstream of the GFP-tag (PMID: 22848743, PMID: 21805523, PMID: 35520093). Nonetheless we have conducted additional experiments designed to estimate the amount of assembled PKD2 and PKD2L1 channels generated using the CFE method. In the new Figure 1— figure supplement 1 D, E, we carried out fluorescencedetection size-exclusion chromatography and compared channel assembly of recombinant and CFE+SUV derived PKD2-GFP and PKD2L1-GFP. Here, we clearly observed tetrameric and protomeric forms of the channels using the synthetic approach, which supports feasibility of using this method to assay their functional properties (see new Figure 1— figure supplement 1 D, E). When compared to protein recombinant sources from HEK cells, the production of assembled channels is less than 4% when using the CFE+SUV approach, an estimate based on the oligomer peak fluorescence.

(4) Single channels were observed more frequently from PKD2 incorporated GUVs compared to PKD2L1. Does this just randomly happen or is there a reason behind this difference?

In terms of overall protein production and oligomeric assembly— more PKD2L1 channels are produced compared to PKD2 (Figure 1C, and Figure 1— figure supplement 1 D, E). This is apparent whether the channels are produced recombinantly in cells or when using the cell-free method (Figure 1— figure supplement 1 D, E). In lines 149-155, we note single channel openings were frequently observed but that the high expression of the PKD2L1 often resulted in patch instability. As a result, GUV patches the lower expressing PKD2-GFP channel were more stable and thus more successfully recorded from. As requested, we have included a brief description of the two proteins in lines 76-78.

(5) Additional validation or clarification for examining the channel orientation may strengthen the manuscript.

We have modified the text to make this point clearer.

(6) Advantage and limitations. The authors compared the recordings from hippocampal primary cilia membranes, noting differences in conductance magnitudes compared to the GUV method. Further discussing the limitations and advantages of this approach for the biophysical properties of organelle channels would be beneficial.

We have revised the final paragraph to discuss the limitations of this method.

(7) Including experiments that demonstrate ligand-induced activation or inhibition to further validate the current using this method would strengthen the manuscript (optional, not required).

Despite our best attempts, exchange of the external bath to apply inhibitors (Gd3+, La3+) resulted in GUV patch instability. Our plans are to investigate ways to stabilize the high resistance seals to develop pharmacological screening using the CFE+GUV method.